# RelGAN: Relational Generative Adversarial Networks for Text Generation

**Weili Nie**[*]
Rice University
wn8@rice.edu

**Nina Narodytska**
VMware Research
nnarodytska@vmware.com

**Ankit B. Patel**
Rice University &
Baylor College of Medicine
abp4@rice.edu

## Abstract

Generative adversarial networks (GANs) have achieved great success at generating realistic images. However, the text generation still remains a challenging task for modern GAN architectures. In this work, we propose RelGAN, a new GAN architecture for text generation, consisting of three main components: a relational memory based generator for the long-distance dependency modeling, the Gumbel-Softmax relaxation for training GANs on discrete data, and multiple embedded representations in the discriminator to provide a more informative signal for the generator updates. Our experiments show that RelGAN outperforms current state-of-the-art models in terms of sample quality and diversity, and we also reveal via ablation studies that each component of RelGAN contributes critically to its performance improvements. Moreover, a key advantage of our method, that distinguishes it from other GANs, is the ability to control the trade-off between sample quality and diversity via the use of a single adjustable parameter. Finally, RelGAN is the first architecture that makes GANs with Gumbel-Softmax relaxation succeed in generating realistic text.

## 1 Introduction

Generative adversarial networks (GANs) (Goodfellow et al., 2014) were originally designed to generate continuous data and have achieved a lot of success at generating continuous samples, such as images. Recently, GANs were extended to generate discrete data, in particular text sequences (Kusner & Hernández-Lobato, 2016; Yu et al., 2017; Zhang et al., 2017; Lin et al., 2017; Guo et al., 2017; Fedus et al., 2018). However, this extension is not straightforward. The main issue is that outputs of GANs for the discrete data generation are not differentiable and thus the standard gradient-based techniques cannot be applied directly in these settings. To overcome this, most state-of-the-art GANs have used the REINFORCE algorithm (Williams, 1992) and its variants that originate from the reinforcement learning (RL) community to train the generator while the discriminator is still a classifier to discriminate real and generated text and provides reward signals for the generator updates. A detailed description of the related work is deferred to Appendix 4.

Although these state-of-the-art GANs have shown some promising results in text generation as compared to the conventional maximum likelihood estimation (MLE) method, they also suffer from some fundamental issues, including training instability and mode collapse. First, their performance is quite sensitive to random parameter initializations and hyperparameter choices (Semeniuta et al., 2018). Moreover, many GANs heavily employ RL heuristics, such as Monte Carlo search (Yu et al., 2017) and hierarchical RL (Guo et al., 2017), making the already difficult-to-train GANs more complicated and thus the individual role of adversarial training unclear. The second issue is mode collapse as the generated text sentences tend to be less diverse (Semeniuta et al., 2018; Fedus et al., 2018) and it becomes more severe when generating longer sentences. The mode collapse issue can be caused either by a lack of expressive power in the generator (since it may not be capable of covering many more complex modes in data distribution), or by a less informative guiding signal in the discriminator (as it may constrain the generator updates to within certain modes).

[*]This work was mostly done during internship at VMware Research. Code for reproducing the core results is available at https://github.com/weilinie/RelGAN.

In this work, we propose a new GAN architecture – Relational GAN (RelGAN), whose design is motivated by the issues identified above. The RelGAN architecture mainly consists of three parts: 1) a *relational memory* (Santoro et al., 2018) based generator, which promises more expressive power and better ability of modeling longer-range dependencies in text; 2) *Gumbel-Softmax relaxation* (Jang et al., 2016; Maddison et al., 2016) for training GANs on discrete data, which simplifies our model, enabling us to stay within a classical GAN framework *without* intensive RL heuristics; 3) *multiple embedded representations* in the discriminator, enabling a more diverse and informative signal for the generator updates. We experimentally demonstrate that RelGAN outperforms most current models in terms of sample quality and diversity. Furthermore, we show via ablation studies that each part of RelGAN plays an important role in its performance improvements. A key advantage of our method, that distinguishes it from other GANs, is the ability to control the trade-off between sample quality and diversity, via the use of a *single adjustable parameter*. Finally, to the best of our knowledge, RelGAN is the first architecture to demonstrate that GANs with Gumbel-Softmax relaxation are capable of generating realistic text.

## 2 RELGAN

### 2.1 RELATIONAL MEMORY BASED GENERATOR

Current dominant GANs for text generation, such as Kusner & Hernández-Lobato (2016); Yu et al. (2017); Lin et al. (2017); Guo et al. (2017); Fedus et al. (2018) are built using LSTM (Hochreiter & Schmidhuber, 1997) as the generator architecture. However, the LSTM-based generator might be the bottleneck of GANs from the following experimental observations: 1) The discriminator's loss value very quickly goes to near its minimum after few iterations of adversarial training. It means that the discriminator may be much more powerful than the generator and can easily distinguish between real and fake samples. 2) Mode collapse in current GANs (Fedus et al., 2018) may also partly indicate the incapacity of generator, as it may not be expressive enough to fit all the modes of data distribution. 3) Current GANs perform poorly at long sentence generation (Guo et al., 2017), and we know that LSTM packs all information about the previous text sequences into a common hidden vector, potentially limiting its ability of modeling the long-distance dependency.

Therefore, we propose to use the more powerful module – relational memory (Santoro et al., 2018) – as the generator architecture for text generation. The basic idea of relational memory is to consider a fixed set of memory slots (e.g. memory matrix) and allow for interactions between memory slots by using the *self-attention* mechanism (Vaswani et al., 2017). The empirical findings by Santoro et al. (2018) showed that relational memory performs better in the language modeling compared to LSTM. Intuitively, the use of multiple memory slots and the attention across these memories can increase the expressive power of generator and its ability of generating longer text sentences.

Formally, we assume each row of the memory $M_t$ represents a memory slot and Figure 1 shows how self-attention updates $M_t$ by incorporating *new observation* $x_t$ at time $t$. Given $H$ heads, we have $H$ sets of queries, keys and values via three linear transformations, respectively: For each head, we get query $Q_t^{(h)} = M_t W_q^{(h)}$, key $K_t^{(h)} = [M_t; x_t] W_k^{(h)}$ and value $V_t^{(h)} = [M_t; x_t] W_v^{(h)}$ where $[;]$ denotes the row-wise concatenation. Thus, the updated memory $\tilde{M}_{t+1}$ is given by

$$\tilde{M}_{t+1} = [\tilde{M}_{t+1}^{(1)} : \cdots : \tilde{M}_{t+1}^{(H)}], \quad \tilde{M}_{t+1}^{(h)} = \sigma \left( \frac{M_t W_q^{(h)} ([M_t; x_t] W_k^{(h)})^T}{\sqrt{d_k}} \right) [M_t; x_t] W_v^{(h)} \tag{1}$$

where $\sigma(\cdot)$ denotes the softmax function which is performed on each row, $d_k$ is the column dimension of the key $K_t^{(h)}$ and $[:]$ denotes the column-wise concatenation.

By following the same idea of Santoro et al. (2018), the next memory $M_{t+1}$ and output (logits) $o_t$ of the generator at time $t$ are given by

$$M_{t+1} = f_{\theta_1}(\tilde{M}_{t+1}, M_t), \quad o_t = f_{\theta_2}(\tilde{M}_{t+1}, M_t) \tag{2}$$

respectively, where the two parametrized functions $f_{\theta_1}$ and $f_{\theta_2}$ are combinations of skip connections, multi-layer perceptron (MLP), gated operations and/or pre-softmax linear transformations.

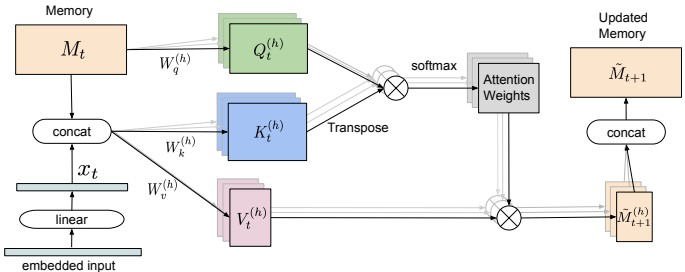

Figure 1: The self-attention mechanism for updating the memory from $M_t$ to $\tilde{M}_{t+1}$ by incorporating new observation $x_t$, where each row of the memory matrix $M_t$ is a memory slot, and $Q_t^{(h)}$, $K_t^{(h)}$ and $V_t^{(h)}$ denote the queries, keys and values, respectively. Note that the softmax function is performed on each row, and $\otimes$ denotes the dot product. The concatenation (denoted by "concat") of $M_t$ and $x_t$ is row-wise where the embedded input is first passed through a linear layer to make $x_t$ match the row dimension of $M_t$.

## 2.2 TRAINING WITH DISCRETE DATA

### 2.2.1 GUMBEL-SOFTMAX RELAXATION

Before the introduction of Gumbel-Softmax relaxation, we first show why training GANs with discrete data is a critical issue. Assuming the vocabulary size is $V$, for the output logits $o_t \in \mathbb{R}^V$ of the generator in (2), the next generated one-hot token $y_{t+1} \in \mathbb{R}^V$ will be obtained by sampling:

$$y_{t+1} \sim \sigma(o_t) \tag{3}$$

where similarly $\sigma(\cdot)$ denotes the softmax function which is performed on $o_t$ element-wisely and we use $\sigma(o_t)$ to represent the multinomial distribution on the set of all possible tokens. As we know, the sampling operations in (3) on the multinomial distribution of the generator output are not differentiable, which implies a step function at the end of the generator. Since the derivative of a step function is 0 almost everywhere, we have $\frac{\partial y_{t+1}}{\partial \theta_G} = 0$ $a.e.$ for $t = 0, \cdots, T-1$ where $\theta_G$ denotes the generator parameters. By chain rule, the gradients of the generator loss $l_G$ w.r.t. $\theta_G$ will be

$$\frac{\partial l_G}{\partial \theta_G} = \sum_{i=0}^{T-1} \frac{\partial y_{t+1}}{\partial \theta_G} \frac{\partial l_G}{\partial y_{t+1}} = 0 \ \ a.e. \tag{4}$$

So the gradients of the generator loss cannot pass back to the generator via the discriminator. This is the notorious "non-differentiability issue" of GANs in discrete data generation.

To deal with the non-differentiablity issue, we apply the Gumbel-Softmax relaxation technique which defines a continuous distribution over the simplex that can approximate samples from a categorical distribution (Jang et al., 2016; Maddison et al., 2016). Formally, the Gumbel-Softmax relaxation includes two parts: 1) The Gumbel-Max trick. According to Jang et al. (2016); Maddison et al. (2016), the sampling in (3) can be reparametrized as

$$y_{t+1} = \text{one\_hot}(\arg\max_{1 \le i \le V}(o_t^{(i)} + g_t^{(i)})) \tag{5}$$

where $o_t^{(i)}$ denotes the $i$-th entry of $o_t$ and $g_t^{(i)}$ is from the $i.i.d.$ standard Gumbel distribution, i.e. $g_t^{(i)} = -\log(-\log U_t^{(i)})$ with $U_t^{(i)} \sim \text{Uniform}(0,1)$. 2) Relaxing the discreteness. As the $\arg\max$ operation in (5) is still non-differentiable, we need further approximate the "one-hot with $\arg\max$" by softmax, which yields

$$\hat{y}_{t+1} = \sigma(\beta(o_t + g_t)) \tag{6}$$

where $\beta > 0$ is a tunable parameter called *inverse temperature*. As $\hat{y}_{t+1}$ in (6) is differentiable with respect to $o_t$, we can use $\hat{y}_{t+1}$ instead of the one-hot token $y_{t+1}$ as the input of the discriminator. Also, note that the *new observation* $x_{t+1}$ of the generator to be concatenated with $M_{t+1}$ in next time $t+1$ is given by $x_{t+1} = f_{\theta_3}(y_{t+1})$, where the parametrized function $f_{\theta_3}$ is composed of an embedding layer that maps $y_{t+1}$ to an embedded input and a linear layer that makes $x_{t+1}$ match the row dimension of $M_{t+1}$ (the embedded input and the linear layer are shown in Figure 1) .

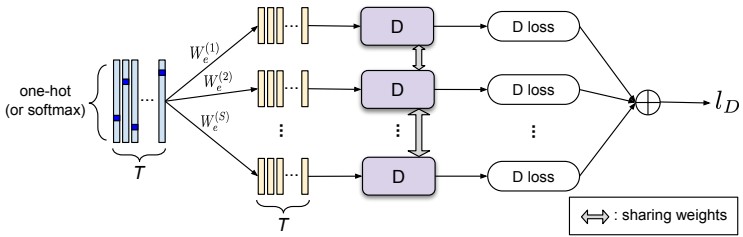

Figure 2: The proposed discriminator framework with multiple embedded representations. The input is either the real sentence $[r_1 : \cdots : r_T]$ where $r_t$ denotes the $t$-th one-hot token, or the generated (approximate) sentence $[\hat{y}_1 : \cdots : \hat{y}_T]$ where $\hat{y}_t$ is from (6). Also, $S$ embedding matrices $\{W_e^{(s)}\}_{s=1}^S$ map each input into $S$ embedded representations, each of which is passed through discriminator independently to get the related loss. Note that "D" is the CNN-based classifier $D(\cdot) \in \mathbb{R}$ with weight-sharing and $\oplus$ denotes taking the average.

### 2.2.2 TEMPERATURE CONTROL

With the larger inverse temperature $\beta$, $\hat{y}_{t+1}$ in (6) will become a better approximation of $y_{t+1}$ in (3) and asymptotically as $\beta \to \infty$, $\hat{y}_{t+1} \to y_{t+1}$. However, the issue is that the variance of gradients will be very large as we have $\text{Var}(\frac{\partial \hat{y}_{t+1}}{\partial o_t}) \propto \beta^2$, and thus the parameter updates will become very sensitive to the input noise. Intuitively, this might cause poor sample quality. On the other hand, with the smaller inverse temperature $\beta$, the generator will pay more attention to making a sharp distribution of entries in $\hat{y}_{i+1}$ due to the larger (initial) approximation gap between $\hat{y}_{t+1}$ and $y_{t+1}$, which implicitly discourages its possible "exploration". Intuitively, this might be one factor that contributes to mode collapse of RelGAN on text generation.

Therefore, the larger $\beta$ encourages more exploration for better sample diversity while the smaller $\beta$ encourages more exploitation for better sample quality. We thus propose to increase the inverse temperature $\beta$ over iterations via an exponential policy: $\beta_n = \beta_{\max}^{n/N}$, where $\beta_{\max}$ denotes the maximum inverse temperature, $N$ is the maximum number of training iterations and $n$ denotes the current iteration. In the exponential policy, as the increase rate of inverse temperature depends on $\beta_{\max}$, $\beta_{\max}$ will decide the transition time from the *exploitation phase* to the *exploration phase*. In such sense, RelGAN provides a flexibility of either generating more diverse samples with a large $\beta_{\max}$ or generating better quality samples with a small $\beta_{\max}$ while most current GANs cannot provide.

### 2.3 MULTIPLE REPRESENTATIONS IN DISCRIMINATOR

A commonly used discriminator for text generation is a CNN-based classifier (Kim, 2014) that employs a convolutional layer with multiple filters of different sizes to capture relations of various word lengths and a max-pooling layer over the entire input sentence for each feature map (Yu et al., 2017; Zhang et al., 2017; Lin et al., 2017; Guo et al., 2017). In this discriminator architecture, the input of the CNN-based classifier is a sentence of length $T$ represented by a *single* embedded matrix $\tilde{X} \in \mathbb{R}^{d \times T}$ where each column $\tilde{x}_t \in \mathbb{R}^d$ is a $d$-dimensional embedded vector of each word.

In this work, we propose a new discriminator framework that applies *multiple* embedded representations for each sentence, with each representation independently passed through the above CNN-based classifier to get an individual score. The average of these individual scores will serve as the final guiding information to update the generator. Our hypothesis is that each embedded representation may capture a specific aspect of the input sentence and the discriminator which compares real and generated sentences from these different perspectives can provide more diverse and comprehensive guiding information for the generator updates. This idea resembles the use of multiple discriminators to improve GANs on image generation (Durugkar et al., 2016), but the difference is that we only use multiple different representations of the input while still keeping a single or weight-sharing CNN-based classifier, which presumably has much less computational cost.

Formally, we assume that $r_t$ denotes the $t$-th one-hot real token and $\hat{y}_t$ from (6) denotes the $t$-th softmax-like generated token, and Figure 2 shows the proposed discriminator framework with multiple embedded representations where either the real input $[r_1 : \cdots : r_T] \in \mathbb{R}^{V \times T}$ or the generated input $[\hat{y}_1 : \cdots : \hat{y}_T] \in \mathbb{R}^{V \times T}$ will be mapped into $S$ embedded representations by $S$ distinct embedding matrices $\{W_e^{(s)}\}_{s=1}^S$ with $W_e^{(s)} \in \mathbb{R}^{d \times V}$. Let $\tilde{X}_r^{(s)}$ and $\tilde{X}_y^{(s)}$ be the $s$-th

embedded representation of the real and generated input, respectively. Thus, we have

$$\tilde{X}_r^{(s)} = W_e^{(s)}[r_1 : \cdots : r_T], \quad \tilde{X}_y^{(s)} = W_e^{(s)}[\hat{y}_1 : \cdots : \hat{y}_T] \tag{7}$$

and the final discriminator loss is given by

$$l_D = \frac{1}{S} \sum_{s=1}^{S} \mathbb{E}_{\substack{r_{1:T} \sim P_r \\ \hat{y}_{1:T} \sim P_\theta}} f(D(\tilde{X}_r^{(s)}), D(\tilde{X}_y^{(s)})) \tag{8}$$

where the expectation is taken w.r.t. both real sentence distribution $P_r$ and generated sentence distribution $P_\theta$, and the loss function $f$ is determined by the specific GAN loss, such as vanilla GAN (Goodfellow et al., 2014), $f$-GAN (Nowozin et al., 2016) and WGAN (Arjovsky et al., 2017). Throughout this paper, the generator loss can be simply set to be $l_G = -l_D$.

## 2.4 TRAINING TECHNIQUES

**Choice of Loss Function.** Empirically, we first compared three different standard GAN losses: standard GAN (the non-saturating version) (Goodfellow et al., 2014), hinge loss (Nowozin et al., 2016; Zhang et al., 2018) and Relativistic standard GAN (RSGAN) (Jolicoeur-Martineau, 2018) on the synthetic data (shown in next section) and then chose the best one – RSGAN for the rest of all experiments. Note that it does not mean RelGAN only works with the RSGAN loss and please see Appendix B for training curves of RelGAN with different loss functions. Formally, the function $f$ in (8) for RSGAN is $f(a, b) = \log \text{sigmoid}(a - b)$ for $a, b \in \mathbb{R}$, and thus (8) becomes

$$l_D = \frac{1}{S} \sum_{s=1}^{S} \mathbb{E}_{\substack{r_{1:T} \sim P_r \\ \hat{y}_{1:T} \sim P_\theta}} \log \text{sigmoid}(D(\tilde{X}_r^{(s)}) - D(\tilde{X}_y^{(s)})) \tag{9}$$

Intuitively, the loss function in (9) is to directly estimate the average probability that real sentences are more realistic than generated sentences in terms of different embedded representations.

**Generator Pre-training.** Most current GANs for text generation need the pre-training for both generator and discriminator, such as SeqGAN (Yu et al., 2017), and some may further heavily rely on the exclusive pre-training techniques, such as TextGAN (Zhang et al., 2017), LeakGAN (Guo et al., 2017) and MaskGAN (Fedus et al., 2018). Instead, the proposed RelGAN only need to pre-train the generator simply via the standard MLE training for several epochs before starting the adversarial training. Experimentally, we find that a good initialization for generator provided by the MLE pre-training is necessary for a good convergence behavior of adversarial training.

## 3 EXPERIMENTS

We test RelGAN on both synthetic and real data, where the synthetic data are 10,000 discrete sequences generated by an oracle-LSTM with fixed parameters (Yu et al., 2017) and the real data include the COCO image captions (Chen et al., 2015) and EMNLP2017 WMT News, first used by Guo et al. (2017) for text generation. The experimental settings are given in Appendix A.

**Evaluation Metrics.** How to properly evaluate generative models remains an open research question (Theis et al., 2015; Semeniuta et al., 2018). The key issue plaging current evaluation metrics for GANs is that they cannot measure *sample quality* and *sample diversity* simultaneously. Therefore, we use two distinct metrics: for synthetic data, we use both negative log-likelihood (called $\text{NLL}_{\text{gen}}$) and its counterpart (called $\text{NLL}_{\text{oracle}}$), defined as:

$$\text{NLL}_{\text{gen}} = -\mathbb{E}_{r_{1:T} \sim P_r} \log P_\theta(r_1, \cdots, r_T), \quad \text{NLL}_{\text{oracle}} = -\mathbb{E}_{y_{1:T} \sim P_\theta} \log P_r(y_1, \cdots, y_T), \tag{10}$$

where the generated sentence distribution $P_\theta$ and the real sentence distribution $P_r$ are both known by evaluating the generator and oracle-LSTM, respectively. Generally, $\text{NLL}_{\text{gen}}$ measures sample diversity while $\text{NLL}_{\text{oracle}}$ is more sensitive to sample quality (Theis et al., 2015; Arjovsky & Bottou, 2017). For the real dataset, we also apply $\text{NLL}_{\text{gen}}$ to measure the sample diversity, similar to Lu et al. (2018). However, since $\text{NLL}_{\text{oracle}}$ cannot be evaluated without an oracle, we instead apply the commonly-used BLEU scores (Papineni et al., 2002) to measure the sample quality and compare with the MLE baseline, along with other start-of-the-art GANs, including SeqGAN (Yu et al., 2017), RankGAN (Lin et al., 2017) and LeakGAN (Guo et al., 2017). Note that for BLEU score evaluation, we follow the strategy in (Yu et al., 2017; Zhu et al., 2018) by using the *test data* as the reference.

### 3.1 SYNTHETIC DATA

We run the synthetic data experiments with sequence length 20 and 40, respectively. The $\text{NLL}_{\text{oracle}}$ results of RelGAN and other models are shown in Table 1 where we set $\beta_{\text{max}} = 1$ for length 20 and $\beta_{\text{max}} = 2$ for length 40. Note that "MLE" in Table 1 denotes the baseline model where LSTMs are trained with the teacher-forcing algorithm to maximize the likelihood (same with Table 2 and 3). We can see that RelGAN outperforms other models in both cases, and its lead in performance becomes larger with longer sequence length, demonstrating the log-distance dependency modeling ability of the proposed generator.

| Length | MLE | SeqGAN | RankGAN | LeakGAN | RelGAN | Real |
|--------|-------|--------|---------|---------|------------------------|-------|
| 20 | 9.038 | 8.736 | 8.247 | 7.038 | **6.680 ± 0.343** | 5.750 |
| 40 | 10.411 | 10.310 | 9.958 | 7.191 | **6.765 ± 0.026** | 4.071 |

Table 1: The $\text{NLL}_{\text{oracle}}$ scores on synthetic data where $\beta_{\text{max}} = 1$ for length 20 and $\beta_{\text{max}} = 2$ for length 40. RelGAN is run with 6 random seeds and the final score is obtained by taking the average of scores, and other scores are from their original papers and Guo et al. (2017). Note that "Real" denotes the real data generated by the oracle-LSTM. For the $\text{NLL}_{\text{oracle}}$ score, the lower the better.

We also evaluate the trade-off between sample quality and diversity as a function of the maximum inverse temperature $\beta_{\text{max}}$ and the results are shown in Figure 3. As $\beta_{\text{max}}$ increases, $\text{NLL}_{\text{gen}}$ decreases, which implies better sample diversity, but $\text{NLL}_{\text{oracle}}$ increases, which implies worse sample quality. Especially when $\beta_{\text{max}} \in \{10, 100\}$, the best $\text{NLL}_{\text{gen}}$ score of 4.4 for RelGAN is very close to the best $\text{NLL}_{\text{gen}}$ score of 4.2 for MLE pre-training, implying that RelGAN with a sufficiently large inverse temperature suffers little mode collapse on synthetic data.

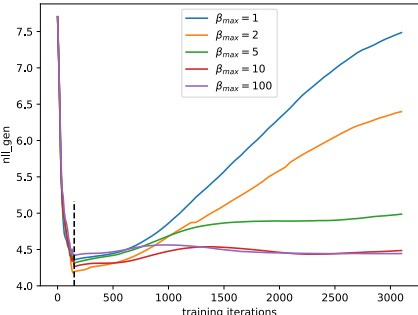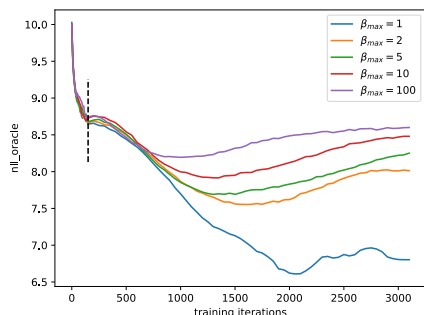

Figure 3: The training curves of $\text{NLL}_{\text{gen}}$ scores (left) and $\text{NLL}_{\text{oracle}}$ scores (right) on synthetic data of length 20 with different values of maximum inverse temperature $\beta_{\text{max}} \in \{1, 2, 5, 10, 100\}$. The vertical dash line represents the end of pre-training. With the increase of $\beta_{\text{max}}$, $\text{NLL}_{\text{gen}}$ becomes lower but $\text{NLL}_{\text{oracle}}$ becomes higher. For both the $\text{NLL}_{\text{gen}}$ and $\text{NLL}_{\text{oracle}}$ scores, the lower the better.

### 3.2 COCO IMAGE CAPTIONS DATASET

In order to test RelGAN on real-world data, we first run experiments on the COCO Image Captions dataset. By following the same data pre-processing as in Zhu et al. (2018), the dataset includes 4,682 unique words with the maximum sentence length 37. Both the training and test data contain 10,000 sentences.

The BLEU scores of RelGAN compared with previous models are shown in Table 2 where we set $\beta_{\text{max}} = 100$ and 1000, respectively. We can see that RelGAN is significantly and consistently better than other models in terms of all the BLEU scores, which means its ability of generating high-quality sentences of COCO Image Captions. Furthermore, the $\text{NLL}_{\text{gen}}$ scores of RelGAN and previous models are also shown in Table 2, where RelGAN also achieves the state-of-the-art results in terms of sample diversity. For example, we do not see obvious mode collapse with $\beta_{\text{max}} = 1000$ by looking at the generated samples (see Appendix C.1 for more details).

| Method | BLEU-2 | BLEU-3 | BLEU-4 | BLEU-5 | NLL$_{\text{gen}}$ |
|---|---|---|---|---|---|
| MLE | 0.731 | 0.497 | 0.305 | 0.189 | 0.718 |
| SeqGAN | 0.745 | 0.498 | 0.294 | 0.180 | 1.082 |
| RankGAN | 0.743 | 0.467 | 0.264 | 0.156 | 1.344 |
| LeakGAN | 0.746 | 0.528 | 0.355 | 0.230 | 0.679 |
| RelGAN (100) | **0.849 ± 0.030** | **0.687 ± 0.047** | **0.502 ± 0.048** | **0.331 ± 0.044** | 0.756 ± 0.054 |
| RelGAN (1000) | 0.814 ± 0.012 | 0.634 ± 0.020 | 0.455 ± 0.023 | 0.303 ± 0.020 | **0.655 ± 0.048** |

Table 2: The BLEU and NLL$_{\text{gen}}$ scores on COCO Image Captions where $\beta_{\max} = 100$ and 1000, respectively. RelGAN is run with 6 random seeds and the final score is obtained by taking the average of scores, and other scores are based on the same evaluation settings in Zhu et al. (2018). For BLEU scores, the higher the better.

## 3.3 EMNLP2017 WMT News Dataset

The EMNLP2017 WMT News dataset consists of 5,255 unique words with the maximum sentence length 51 after applying the same data pre-processing as in Zhu et al. (2018). The training data contains abbout 270,000 sentences and test data contains 10,000 sentences.

| Method | BLEU-2 | BLEU-3 | BLEU-4 | BLEU-5 | NLL$_{\text{gen}}$ |
|---|---|---|---|---|---|
| MLE | 0.768 | 0.473 | 0.240 | 0.126 | 2.382 |
| SeqGAN | 0.777 | 0.491 | 0.261 | 0.138 | 2.773 |
| RankGAN | 0.727 | 0.435 | 0.209 | 0.101 | 3.345 |
| LeakGAN | 0.826 | 0.645 | 0.437 | 0.272 | 2.356 |
| RelGAN (100) | **0.881 ± 0.013** | **0.705 ± 0.019** | **0.501 ± 0.023** | **0.319 ± 0.018** | 2.482 ± 0.031 |
| RelGAN (1000) | 0.837 ± 0.012 | 0.654 ± 0.010 | 0.435 ± 0.011 | 0.265 ± 0.011 | **2.285 ± 0.025** |

Table 3: The BLEU and NLL$_{\text{gen}}$ scores on EMNLP2017 WMT News where $\beta_{\max} = 100$ and 1000, respectively. Our model is run with 6 random seeds and the final score is obtained by taking the average of scores, and other scores are based on the same evaluation settings in Zhu et al. (2018).

The BLEU scores of RelGAN compared with previous models are shown in Table 3 where we set $\beta_{\max} = 100$ and 1000, respectively. We can see that RelGAN also consistently outperforms previous models in terms of all the BLEU scores, demonstrating its ability of generating high-quality sentences on EMNLP2017 WMT News. Moreover, the sample diversity metric NLL$_{\text{gen}}$ scores of RelGAN and previous models are also shown in Table 3. Similarly, RelGAN achieves the state-of-the-art results in terms of sample diversity. Upon visually examining generated samples (See Appendix C.2 for more details), we do not observe obvious mode collapse for $\beta_{\max} \in \{100, 1000\}$.

Finally, from Tables 2 and 3, we can see that the sample quality and diversity trade-off with different values of maximum inverse temperature $\beta_{\max}$ also exists on the real data. That is, RelGAN with $\beta_{\max} = 100$ achieves better sample quality while RelGAN with $\beta_{\max} = 1000$ achieves better sample diversity. Depending on what the underlying applications of text generation via RelGAN are, we can adjust $\beta_{\max}$ properly to get either better quality or better diversity.

To further evaluate the sample quality of RelGAN and other models on EMNLP2017 WMT News, we also perform the human evaluation by using Amazon Mechanical Turk. We randomly sampled 100 sentences for each model and the real dataset, and asked 10 different people to score each sentence on a scale of 1-5. Please see Appendix A.2 for more details of human evaluation.

| Method | MLE | SeqGAN | RankGAN | LeakGAN |
|---|---|---|---|---|
| Human score | 2.751 ± 0.908 | 2.588 ± 0.970 | 2.449 ± 1.051 | 3.011 ± 0.908 |
| Method | RelGAN(100) | RelGAN(1000) | Real | |
| Human score | **3.407 ± 0.909** | 3.285 ± 0.900 | 4.445 ± 0.679 | |

Table 4: The means and standard deviations of human scores for RelGAN and other models on EMNLP2017 WMT News by using Amazon Mechanical Turk. Note that "Real" denotes the human score on the real dataset.

The human score results are provided in Table 4, where we can see that RelAGN generates better human-looking samples than other GANs and the MLE baseline model.

### 3.4 ABLATION STUDY

#### 3.4.1 IMPACT OF RELATIONAL MEMORY

To show the impact of relational memory in RelGAN, we propose to replace relational memory by LSTM-32 and LSTM-512 as the generator architecture, respectively, and see how the performance differs. Here LSTM-$k$ represents the LSTM with hidden dimension being $k$. We choose $k = 32$ because most previous GANs (Yu et al., 2017; Guo et al., 2017) have used this architecture for text generation, and also choose $k = 512$ because for more fair comparison, we want to keep the total memory size of LSTM to be the same with the relational memory we have used.

The results on the COCO Image Captions dataset are shown in Figure 4 (Left), where we provide the BLEU-4 score (See Appendix D.1 for all the BLEU scores). We can see that the BLEU scores of relational memory are consistently better than those of LSTM-32 and LSTM-512, which demonstrates the advantages of using relational memory as generator in RelGAN.

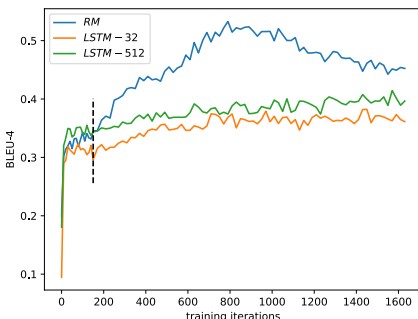 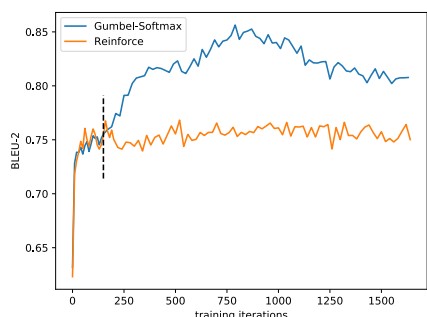

Figure 4: (Left) Training curves of the BLEU-4 score on COCO Image Captions with different generator architectures – relational memory (RM), LSTM-32 and LSTM-512. (Right) Training curves of the BLEU-2 score on COCO Image Captions with Gumbel-Softmax relaxation and the vanilla REINFORCE method. All the results are obtained by taking the average of 6 runs with different random seeds.

#### 3.4.2 IMPACT OF GUMBEL-SOFTMAX RELAXATION

To show the impact of Gumbel-Softmax relaxation in RelGAN, we can instead apply the vanilla REINFORCE method to deal with the non-differentiable issue of RelGAN on text generation. In this experiment, we keep all other hyperparameters in RelGAN fixed and compare the performance of Gumbel-Softmax relaxation and the vanilla REINFORCE method.

The results are shown in Figure 4 (Right), where the BLEU-2 score is provided (See Appendix D.2 for all the BLEU scores). We can see that under the proposed RelGAN framework, Gumbel-Softmax relaxation performs much better than the vanilla REINFORCE method. During experiments, we find that the variance of generator gradients in the vanilla REINFORCE method is too large to provide any useful update for generator, which may explain why the performance of vanilla REINFORCE does not improve after the pre-training, as observed in Figure 4 (Right). The exploration of various variance reduction techniques for the REINFORCE method in RelGAN is out of scope of this paper.

#### 3.4.3 IMPACT OF MULTIPLE REPRESENTATIONS IN DISCRIMINATOR

To show the impact of multiple embedded representations in discriminator while keeping the expressive power of discriminator the same for fair comparison, we propose to apply $S$ embedded presentations with each embedded vector of length $d = \frac{d_{\max}}{S}$ where $d_{\max}$ denotes the total length of representations. In this experiment, we set $d_{\max} = 64$, and thus for instance, if $S = 1$ then $d = 64$ for each embedded vector, and if $S = 2$ then $d = 32$ for each embedded vector, and so on.

We first test RelGAN on the synthetic data with $S \in \{1, 2, 4, 8, 16, 32, 64\}$ and the results are shown in Figure 5 (Left). We can see that as the number of embedded representations $S$ increases, the best NLL$_{\text{oracle}}$ score tends to keep decreasing, yielding better sample quality. Furthermore, we test RelGAN on COCO Image Captions with $S \in \{1, 64\}$ and the BLEU-3 score is shown in Figure 5 (Right). Still, we can see that the BLEU scores of RelGAN with $S = 64$ are consistently better than those of RelGAN with $S = 1$ (see Appendix D.3 for all the BLEU scores). Note that in both experi-

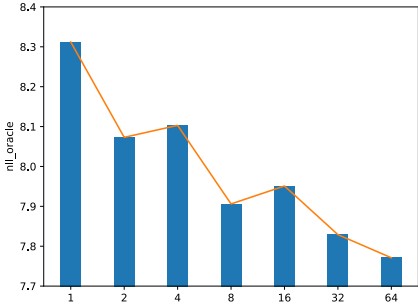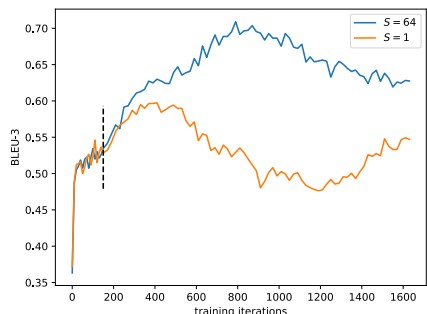

Figure 5: (Left) The best NLL$_{\text{oracle}}$ score on the synthetic data varies with different number of embedded presentations $S = \{1, 2, 4, 8, 16, 32, 64\}$ where $\beta_{\max} = 10$. (Right) The training curves of BLEU-3 score on COCO Image Captions with the number of embedded representations $S = 1$ and $S = 64$, respectively, where $\beta_{\max} = 1000$. All results are obtained by taking the average of 6 runs with different random seeds.

ments, we do not see an obvious sign of mode collapse with varying number of representations. For example, the NLL$_{\text{gen}}$ score on the synthetic data stays around $4.4$ for different values of $S$ (close to the best NLL$_{\text{gen}}$ $4.2$ for MLE shown in Figure 3 (Left)). Thus, these experiments demonstrate the advantages of using multiple embedded representations for discriminator in RelGAN.

## 4 RELATED WORK

Since GANs are originally proposed for continuous data, extending GAN training to discrete data generation has been an active research topic. Current works focus on dealing with the non-diferentiable issue brought by the discrete data nature either by considering the RL methods or by reformulating the problem in continuous space.

A large class of GANs for text generation relies on the RL algorithm. SeqGAN ((Yu et al., 2017)) models the text generation as a sequential decision making process and trains the generator with policy gradient methods (Sutton et al., 2000). MaliGAN (Che et al., 2017) proposes the co-training with a maximum-likelihood objective to reduce the gradient variance. RankGAN (Lin et al., 2017) proposes a ranking model to replace the original binary classifier as the discriminator. LeakGAN (Guo et al., 2017) designs a mechanism to provide intermediate information about text generation for generator, where the discriminator can leak its features through a manager module. MaskGAN (Fedus et al., 2018) introduces an actor-critic conditional GAN that fills in missing text conditioned on the surrounding context by resorting to a seq2seq model.

Other GANs without RL methods either approximate the discrete data or work in the continuous latent space. TextGAN (Zhang et al., 2017) provides a feature matching mechanism that matches the latent features of real and generated sentences via a kernelized discrepancy metric to alleviate the mode collapse. FM-GAN (Chen et al., 2018) proposes to match the latent feature distributions of real and synthetic sentences using the feature-movers distance. Similar to our work, both textGAN and FM-GAN apply an annealed softmax to approximate the argmax in the generator. However, they do not rely on the Gumbel-Max trick to reparametrize the sampling operations, which is the major difference with us in dealing with the non-differentiable issue. ARAE (Zhao et al., 2018) applies an additional autoencoder to embed the discrete data into a continuous latent space in which GANs can be trained properly. As for approximating the categorical distribution with Gumbel-Softmax relaxation, Gu et al. (2017) has used it to improve the generation quality in neural machine translation. More similarly, Kusner & Hernández-Lobato (2016) provides some initial experiments of training GANs with Gumbel-Softmax relaxation on a synthetic task, but scaling them to work on real text dataset remains a challenging open problem.

Attention mechanisms, in particular self-attention (Vaswani et al., 2017), have gradually become a building block of many novel neural network architectures (Vaswani et al., 2017; Parmar et al., 2018; Santoro et al., 2018) due to its ability of capturing long or global dependencies and reducing computational cost via parallelization. In the context of GANs, self-attention have not been fully explored. SAGAN (Zhang et al., 2018) applies self-attention in GANs to model long range depen-

dencies in images and get the state-of-the-art results on conditional image generation. In contrast, we employ self-attention in GANs for text generation by using relational memory as generator.

## 5 CONCLUSIONS

We proposed a new GAN architecture called RelGAN for text generation, that outperforms most current models in terms of sample quality and diversity on both synthetic and real data. Furthermore, the trade-off between the generated sample diversity and quality can be adjusted properly in RelGAN by controlling the inverse temperature. In RelGAN, we used the relational memory based generator to improve its ability of modeling long distance dependencies and also applied multiple embedded representations in discriminator such that it can provide more diverse and informative guiding signal for generator. By applying Gumbel-Softmax relaxation to deal with the non-differentiable issue, our architecture is simple to implement without employing intensive RL heuristics. For the future directions, since we have demonstrated that GANs with Gumbel-Softmax relaxation is very promising for text generation, we would like to explore further in this direction. For example, it is interesting to make RelGAN work better without any pre-training. Also, extending RelGAN to a conditional model for many text generation related applications is another interesting direction.

## ACKNOWLEDGEMENT

We would like to thank all the reviewers for their helpful comments. WN and ABP were supported by IARPA via DoI/IBC contract D16PC00003 and NSF NeuroNex grant DBI-1707400.

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

## A  EXPERIMENTAL SETTINGS

### A.1  TRAINING DETAILS

Unless stated otherwise, for the CNN-based discriminator architecture, we use filter windows of sizes {3,4,5} and 300 feature maps each. For relational memory, we set memory size to be 256, memory slots to be 1, number of heads to be 2. The batch size is set to be 64. For embedding dimensions, we set the embedding dimension of the input token for generator to be 32 and that for discriminator to be 1 with the number of embedded representations $S = 64$. We use Adam (Kingma & Ba, 2014) with $\beta_1 = 0.9$ and $\beta_2 = 0.999$ and gradient clipping is applied if the norm of gradients exceeds 5. We first pre-train the generator via MLE with learning rate of 1e-2 for 150 epochs and then start adversarial training with learning rate of 1e-4 for both discriminator and generator. For adversarial training, we set the maximum number of iterations $N = 5000$ and we perform 5 gradient descent steps on the discriminator for every step on the generator.

### A.2  HUMAN EVALUATION DETAILS

On Amazon Mechanical Turk, our instructions are given as follows:

The text quality evaluation is based on grammatical correctness and meaningfulness (i.e. if a sentence makes sense). Please ignore any text formatting problems (e.g., capitalization, punctuation, spelling errors, extra spaces between words and punctuations). Note: A very short sentence (less than 10 words) should be penalized with its score minus 1. Please see below for the detailed criteria.

| Scale | Criterion & Example |
|---|---|
| 5 - Excellent | Its grammatically correct and makes sense. For example, "*if England wins the World Cup next year , it will be the most significant result the sport has seen in more than a decade .*" |
| 4 - Good | It has some small grammatical errors and mostly makes sense. For example, "*it is useful to have had a doctor who forced her to release him a couple of days before she was cleared .*" |
| 3 - Fair | It has major grammatical errors but the whole still conveys some meanings. For example, "*even then once again there ' s a sign of that stuff is going on the way to work on christmas eve .*" |
| 2 - Poor | It has severe grammatical errors and the whole doesn't make sense, but some parts are still locally meaningful. For example, "*we go to work for the moment in life their eyes and , i have been a different race on to go .*" |
| 1 - Unacceptable | It is basically a random collection of words. For example, "*i go com com com , i on on on play can go go .*" |

Table 5: The human evaluation scale from 1 to 5 with corresponding criteria and example sentences.

# B  RELGAN WITH VARYING HYPERPARAMETERS

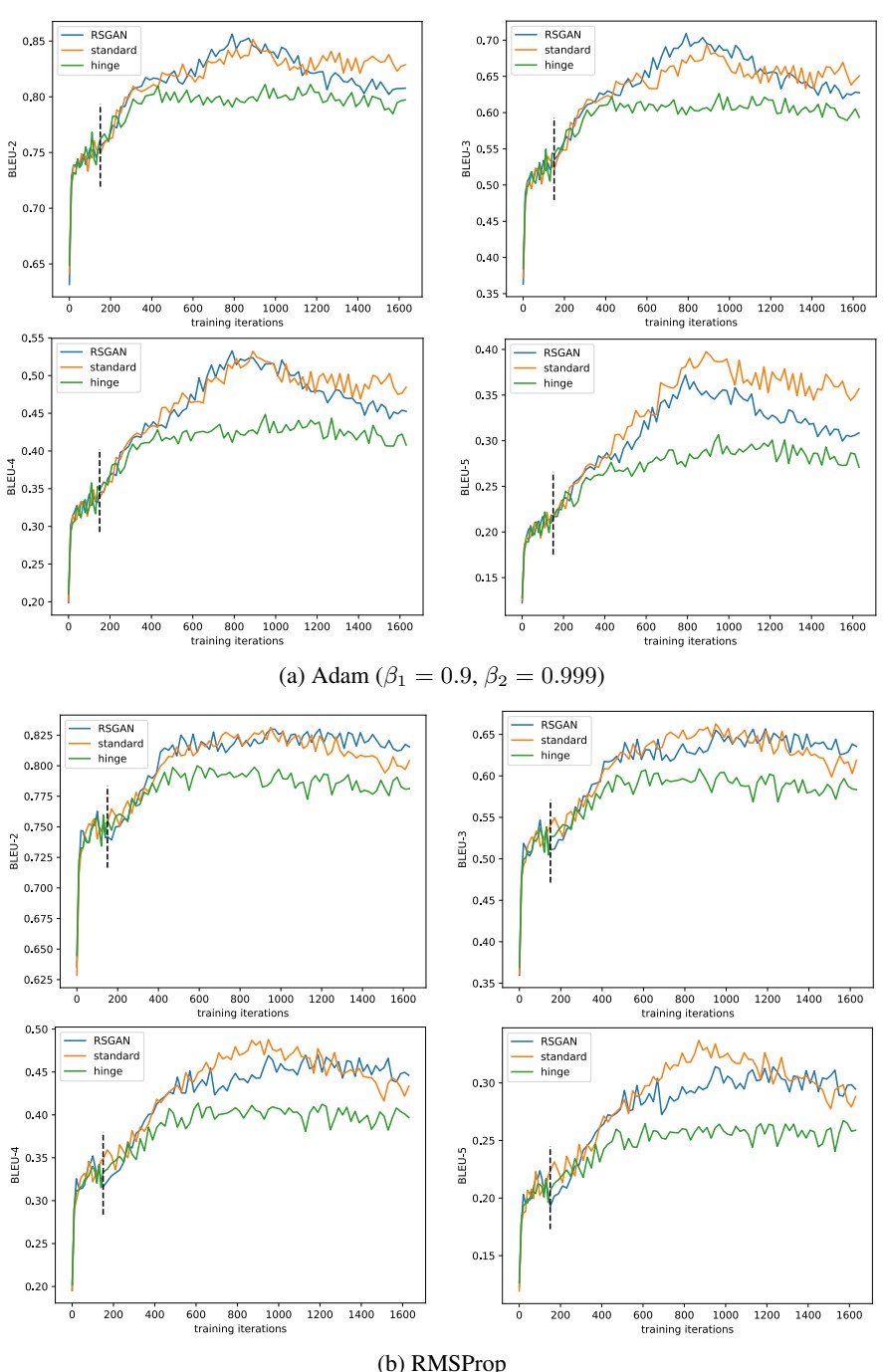

(a) Adam ($\beta_1 = 0.9$, $\beta_2 = 0.999$)

(b) RMSProp

Figure 6: Training curves of BLEU scores on COCO Image Captions with different loss functions: RSGAN (Jolicoeur-Martineau, 2018), standard GAN (the non-saturating version) (Goodfellow et al., 2014) and hinge loss (Nowozin et al., 2016; Zhang et al., 2018), where $\beta_{\max} = 1000$ and we use two different optimizers – (a) Adam and (b) RMSProp. All the results are obtained by taking the average of 6 runs with different random seeds. The vertical dash line represents the end of pre-training. We can see that RelGAN works well with different commonly-used loss functions of GANs and different optimization methods. In this scenario, the performance of RSGAN and standard GAN outperforms the hinge loss version.

# C    GENERATED SAMPLES ON REAL DATASET

## C.1    GENERATED SAMPLES ON COCO IMAGE CAPTIONS DATASET

---

a man is sitting on a bench next to a bicycle .

a man fixing a motor cycle in a race .

a rectangle shaped wooden sitting on a lush green field .

a man standing in a picture of a kitchen with a dog watching him .

a man is carving some meat in a park .

a kitchen with a black window and a large white stove wall .

a cat is looking on a man in a bathroom .

a train is covered in the air in the city scene .

a bathroom has a toilet , and bathroom rug for a urinal or a urinal on the side .

a home kitchen with a double oven and table while a table and chairs .

a small airplane flying above an airport covered with wood .

a man in a kitchen preparing food on a table .

a large passenger jet flying in a clear blue sky .

a group of people riding motorcycles on a street .

a woman in a kitchen with her hands clasped .

---

a smiling woman is sitting on a green bench .

people are hiding under colorful umbrellas on a rainy day .

a photo of a small restroom in a kitchen .

many sheep graze are shown in front of a group of people .

a plane is parked next to an airplane on a runway near a control tower with two back .

a woman walking past a straw shower holding a corner .

an office desk with a row of books in the kitchen .

a cat sitting on top of a kitchen counter .

a person riding a bike through a lush green park .

a bathroom with a sink , toilet and toilet paper dispenser .

a man is on a motorcycle with a woman on the back of it .

two giraffes in a wild , lightly wooded field .

a metal tin pan filled with two different kitchen appliances .

a white airplane flying in the sky over a runway .

a car driving on a busy street at night from an airport .

---

Table 6: Randomly chosen 15 generated samples trained on COCO image caption data with $\beta_{max} = 100$ (top row) and $\beta_{max} = 1000$ (bottom row). We can see that for $\beta_{max} = 100$, the words "man" and "kitchen" occur severe times even in these small set of generated sentences, meaning some sort of lacking sentence diversity. For $\beta_{max} = 1000$, however, there is no sign of obviously poor sentence diversity by human eyes.

## C.2   Generated Samples on EMNLP2017 WMT News Dataset

he is also watching closely on staying in the plan , but sometimes i still don ' t think that ' s going to happen on saturday .

and so , they didn ' t want to be in their position , and that was something they would do to my wife .

i would like to assess whether you should be able to take that into that issue than you did on a sunday .

" he ' s a young lady , so i don ' t want to show up for them , " she said .

that ' s not yet about what you want to say about reality , but probably don ' t think about that would just make me comfortable in my life .

officials have been also a member of nato against the us , and for the first time that so many other countries are on the rise , and that will be the only way to stay here .

he has always vowed to give him a little more on for him , and i think he is very willing for the division .

we ' ve had to try and get into the coming down to the today ' s end .

i don ' t think that ' s why we did not have to score the last two .

meanwhile , it was never been in the past , but it was not known until the coalition was given the support of the rebels on the terms of the claims .

he said : " i don ' t think we should be better at what we would do to the good .

" we had to get on with that , " she said at a news conference .

by contrast , but this is a turning point at her age .

but i will do that , which i have to do with the city , and my new hopes is from all over the world .

" this is a very bad , and it ' s not clear that this is a danger of this , " he said .

at a detroit press with reporters on the flight , gave informed operators time to interview mr . cox that he was struggling with all of his treatments .

" in the past five days , we ' re going to enjoy maybe that and after that , you ' re going to stay to dinner with friends and family , " he said .

the union has indicated that there are no restrictions on the long - standing alliance in any key areas .

i never thought i could put over the line but i couldn ' t quite lose my job .

since then , while the number of people stood by the wall street banks fell by 2 percent over the past 10 years , there ' s no need to say that .

" they had to fly in the field , " he said , adding that it didn ' t miss it .

" trump ' s voice will be a positive one for mr . trump ' s transition team , " he said .

he was still working on a training camp friday with a small into a new manager .

the 15 - year - old man has been reported missing by a falling from the city centre .

" it ' s a process that can take a little while , " one resident said .

a well - meaning - predicted or very public policy , seeking to work with us .

" i ' m not to have made me a bit more than anything , but i ' ve never done that , " he said .

they were waiting to see how many changes could come from us and that ' s why it has made it leave facing .

she initially noted that some of the other victims began to come from being more than prepared to stand for .

" i ' ve never heard of the abuse , because we need to work with him , " he added .

Table 7: Randomly chosen 15 generated samples trained on EMNLP WMT News dataset with $\beta_{max} = 100$ (top row) and $\beta_{max} = 1000$ (bottom row). We can see that in both cases, there is no obvious sign of poor sentence diversity by human eyes.

# D MORE RESULTS ON ABLATION STUDY

## D.1 IMPACT OF RELATIONAL MEMORY

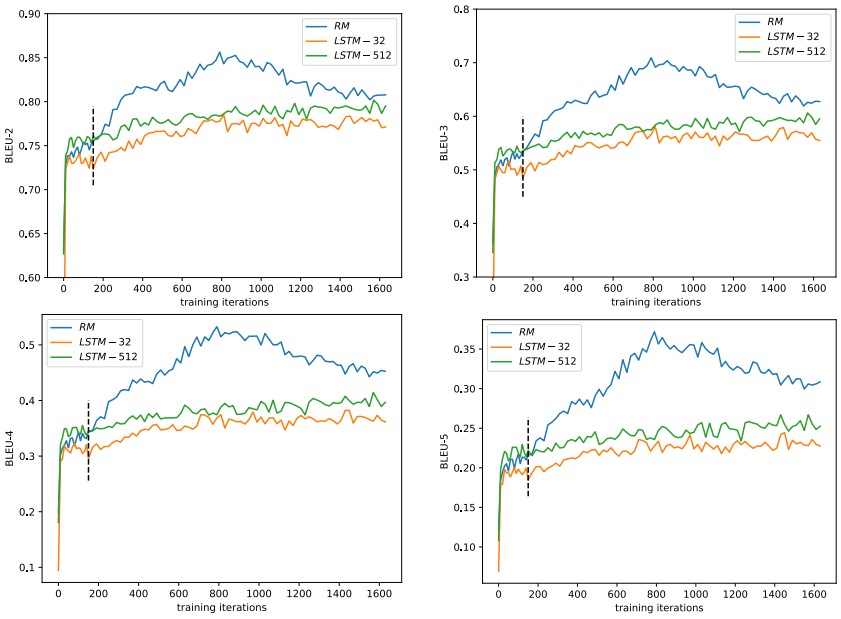

Figure 7: Training curves of BLEU scores on COCO Image Captions with different generator architectures – relational memory (RM), LSTM-32 and LSTM-512, where $\beta_{\max} = 1000$. We can see that the BLEU scores of relational memory are consistently better than those of LSTM-32 and LSTM-512, which demonstrates the advantages of using relational memory as generator in RelGAN.

## D.2 IMPACT OF GUMBEL-SOFTMAX RELAXATION

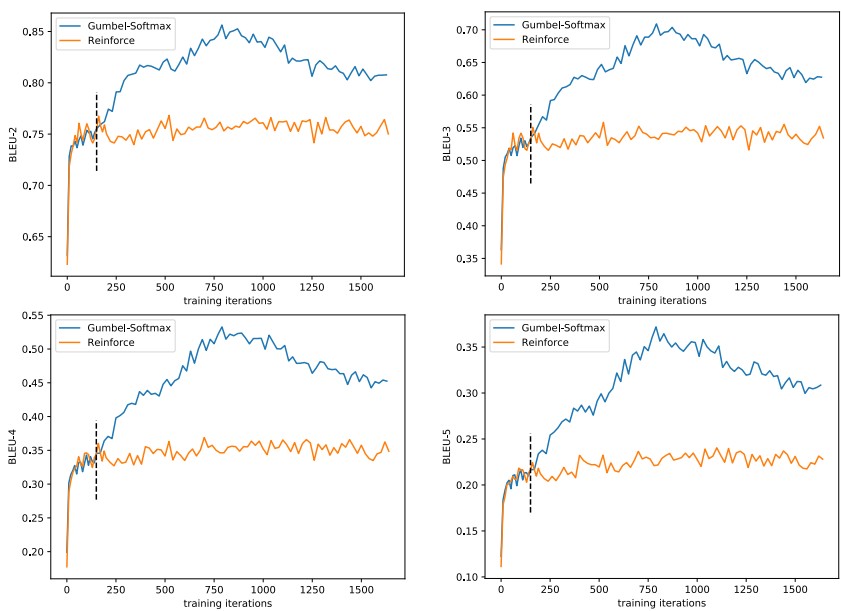

Figure 8: Training curves of BLEU scores on COCO Image Captions with different gradient relaxations for GANs on discrete data – Gumbel-Softmax relaxation and REINFORCE method. We can see that the BLEU scores of Gumbel-Softmax relaxation are consistently better than those of REINFORCE method, which demonstrates the advantages of using Gumbel-Softmax relaxation to deal with non-differentiable issues in RelGAN.

### D.3 IMPACT OF MULTIPLE REPRESENTATIONS FOR DISCRIMINATOR

Figure 9: Training curves of BLEU scores on COCO Image Captions with different number of embedded representations $S = 1$ and $S = 64$, where $\beta_{\max} = 1000$. We can see that the BLEU scores of $S = 64$ are consistently better than those of $S = 1$, which demonstrates the advantages of using multiple embedded representations for discriminator in RelGAN.

## E  DIVERSITY-QUALITY TRANSITION DURING TRAINING

As we can observe from Figures 7-9 in the above appendices, the BLEU scores of RelGAN (denoted by the blue curves in each subfigure) first increase over iterations and then keep decreasing after around 800 iterations. In other words, its sample quality first increases and then decreases during the adversarial training. To see what happens in the training dynamics of RelGAN, we also provide the training curve of the diversity metric – $\text{NLL}_{\text{gen}}$ in RelGAN as shown in Figure 10.

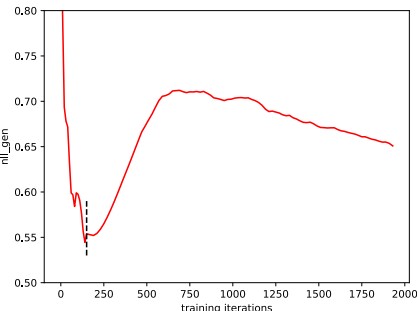

Figure 10: The training curve of $\text{NLL}_{\text{gen}}$ in RelGAN on COCO Image Captions, where $\beta_{\max} = 1000$. We can see that during the adversarial training, the $\text{NLL}_{\text{gen}}$ score first increases and then decreases after around 800 iterations. The turning point matches well with those in the training curves of BLEU scores.

Interestingly, Figure 10 shows that during the adversarial training, the sample diversity (measured by $\text{NLL}_{\text{gen}}$) of RelGAN first decreases and then increases, and its turning point matches well with that of the sample quality (measured by BLEU scores) of RelGAN. These training dynamics illustrate a diversity-quality transition over iterations in RelGAN: Early on in training, it learns to aggressively improve sample quality while sacrificing diversity. Later on, it turns instead to maximizing sample

diversity while gradually decreasing sample quality. Intuitively, it seems to be much easier for the generator to just produce realistic samples – regardless of their diversity – to fool the discriminator in the early stage of training. As the discriminator becomes better at distinguishing samples with less diversity over iterations, the generator has to focus more on producing more diverse samples to fool the discriminator.

## F    RELGAN WITHOUT PRE-TRAINING

In this section, we want to test the performance of RelGAN *without pre-training* for different loss functions, including standard GAN (the non-satuarating version) (Goodfellow et al., 2014), WGAN-GP (Gulrajani et al., 2017), hinge loss (Nowozin et al., 2016; Zhang et al., 2018) and RSGAN (Jolicoeur-Martineau, 2018). The BLEU and $\text{NLL}_{\text{gen}}$ scores evaluated on COCO Image Captions are shown in Table 8.

| Losses | BLEU-2 | BLEU-3 | BLEU-4 | BLEU-5 | $\text{NLL}_{\text{gen}}$ |
|---|---|---|---|---|---|
| Hinge | - | - | - | - | - |
| WGAN-GP | $0.330 \pm 0.024$ | $0.111 \pm 0.019$ | $0.065 \pm 0.017$ | $0.045 \pm 0.013$ | $4.063 \pm 0.623$ |
| RSGAN | $0.460 \pm 0.026$ | $0.172 \pm 0.025$ | $0.085 \pm 0.017$ | $0.056 \pm 0.013$ | $3.065 \pm 0.917$ |
| Standard | $\mathbf{0.590 \pm 0.019}$ | $\mathbf{0.280 \pm 0.020}$ | $\mathbf{0.141 \pm 0.018}$ | $\mathbf{0.094 \pm 0.011}$ | $\mathbf{2.259 \pm 0.263}$ |
| Random | 0.041 | 0.017 | 0.011 | 0.008 | 8.355 |

Table 8: The BLEU and $\text{NLL}_{\text{gen}}$ scores of RelGAN without pre-training on COCO Image Captions where with a little hyperparameter tuning, we set $\beta_{\max} = 100$ for the standard GAN loss and $\beta_{\max} = 1000$ for other losses. All the results are run with 6 random seeds and the final score is obtained by taking the average of scores. As a reference, we also provide the results of an untrained RelGAN which is marked as "random". Note that for the hinge loss, we get no valid results as it suffers from the vanishing gradient issue.

We can see that without pre-training, there is still a significant improvement for RelGAN compared with the random generation, in particular for the standard GAN loss, even though the improvement is inferior to the case with pre-training. In contrast, without pre-training, previous RL-based GANs for text generation, such as SeqGAN and RankGAN, always get stuck around their initialization points and are not able to improve their performance at all. This demonstrates that RelGAN may be a more promising GAN architecture to explore in order to completely get rid of the pre-training for GANs on text generation. Besides, Table 8 also shows that the evaluation results of RelGAN without pre-training vary with different loss functions and values of $\beta_{\max}$. We leave an extensive hyperparameter search to further improve the performance of RelGAN without pre-training for future work.

## G    MORE EXPLORATION ON TUNABLE HYPERPARAMETER $\beta_{\max}$

For real data experiments, we have showed the advantages of RelGAN over other models by setting the maximum inverse temperature $\beta_{\max} \in \{100, 1000\}$, which are carefully chosen for a good trade-off between sample quality and diversity. A natural question will be to explore the two extremes: what happens with the real data if $\beta_{\max}$ is either too large or too small? Does it behave similarly to the synthetic data experiments in terms of the trade-off between sample diversity and quality?

To this end, we choose a broad range of $\beta_{\max} \in \{1, 10, 10^2, 10^3, 10^4, 10^5, 10^6, 10^7\}$ and test its impact in RelGAN on the COCO Image Captions dataset. The BLEU and $\text{NLL}_{\text{gen}}$ scores are given in Table 9, where we can see that both BLEU and $\text{NLL}_{\text{gen}}$ scores increase with the decrease of $\beta_{\max}$, and the variance of each score also consistently becomes larger for a smaller $\beta_{\max}$. For better illustration, we also plot BLEU-4 and $\text{NLL}_{\text{gen}}$ scores with error bars in Figure 11.

It first confirms that similar to the synthetic data experiments, there also exists a consistent trade-off between sample quality and diversity in real data, controlled by the tunable hyperparameter $\beta_{\max}$. Besides, it reveals the failing cases at the two extremes: On the one hand, if $\beta_{\max}$ is too small, i.e. $\beta_{\max} = 1$, RelGAN suffers from severe mode collapse (denoted by the large $\text{NLL}_{\text{gen}}$) and training instability (denoted by high variances of scores) issues. On the other hand, if $\beta_{\max}$ is too large, i.e. $\beta_{\max} = 10^7$, the sample quality improvement of RelGAN becomes marginal (denoted by the low BLEU scores). Therefore, we have chosen the two intermediate values $\{100, 1000\}$ of $\beta_{\max}$ in the

| $\beta_{\max}$ | BLEU-2 | BLEU-3 | BLEU-4 | BLEU-5 | $\text{NLL}_{\text{gen}}$ |
|---|---|---|---|---|---|
| 1 | $\mathbf{0.890 \pm 0.121}$ | $\mathbf{0.791 \pm 0.209}$ | $\mathbf{0.659 \pm 0.243}$ | $\mathbf{0.500 \pm 0.230}$ | $1.454 \pm 0.121$ |
| 10 | $0.862 \pm 0.038$ | $0.741 \pm 0.060$ | $0.604 \pm 0.060$ | $0.445 \pm 0.080$ | $1.084 \pm 0.061$ |
| $10^2$ | $0.849 \pm 0.030$ | $0.687 \pm 0.047$ | $0.502 \pm 0.048$ | $0.331 \pm 0.044$ | $0.756 \pm 0.054$ |
| $10^3$ | $0.814 \pm 0.012$ | $0.634 \pm 0.020$ | $0.455 \pm 0.023$ | $0.303 \pm 0.020$ | $0.655 \pm 0.048$ |
| $10^4$ | $0.801 \pm 0.006$ | $0.609 \pm 0.012$ | $0.430 \pm 0.019$ | $0.288 \pm 0.015$ | $0.631 \pm 0.045$ |
| $10^5$ | $0.796 \pm 0.007$ | $0.599 \pm 0.012$ | $0.417 \pm 0.010$ | $0.277 \pm 0.012$ | $0.588 \pm 0.037$ |
| $10^6$ | $0.790 \pm 0.009$ | $0.588 \pm 0.011$ | $0.408 \pm 0.013$ | $0.272 \pm 0.010$ | $0.569 \pm 0.039$ |
| $10^7$ | $0.775 \pm 0.011$ | $0.572 \pm 0.020$ | $0.390 \pm 0.019$ | $0.252 \pm 0.016$ | $\mathbf{0.547 \pm 0.032}$ |

Table 9: The BLEU and $\text{NLL}_{\text{gen}}$ scores of RelGAN with various values of $\beta_{\max}$ on COCO Image Captions. All the results are run with 6 random seeds and the final score is obtained by taking the average of scores. As we can see, both BLEU and $\text{NLL}_{\text{gen}}$ scores increase with the decrease of $\beta_{\max}$. Besides, the variance of each score also consistently becomes larger for a smaller $\beta_{\max}$.

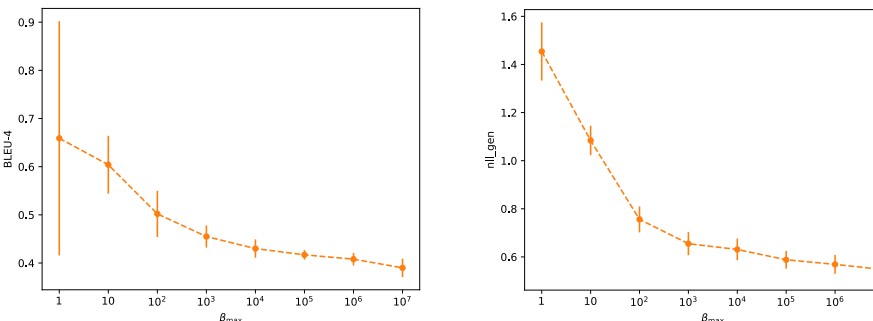

Figure 11: The BLEU-4 (Left) and $\text{NLL}_{\text{gen}}$ (Right) scores with error bars in RelGAN on COCO Image Captions with varying maximum inverse temperature $\beta_{\max}$.

main text to show the advantages of RelGAN over other models while still demonstrating its ability to control the trade-off between sample quality and diversity.

