# OpenReview forum: "RelGAN: Relational Generative Adversarial Networks for Text Generation"
_ICLR.cc/2019/Conference_

### Official Review · AnonReviewer3 · 2018-11-02
**interesting idea and experiments well-executed**

**Rating:** 6
**Confidence:** 4

**Review:**

==========================
I have read the authors' response and other reviewers' comments. Thanks the authors for taking great effort in answering my questions. Generally, I feel satisfied with the repsonse, and prefer an acceptance recommendation.
==========================
Contributions:

The main contribution of this paper is the proposed RelGAN. First, instead of using a standard LSTM as generator, the authors propose using a relational memory based generator. Second, instead of using a single CNN as discriminator, the authors use multiple embedded representations. Third, Gumbel-softmax relaxation is also used for training GANs on discrete textual data. The authors also claim the proposed model has the ability to control the trade-off between sample quality and diversity via a single adjustable parameter.

Detailed comments:

(1) Novelty: This paper is not a breakthrough paper, mainly following previous work and propose new designs to improve the performance. However, it still contains some novelty inside, for example, the model choice of the generator and discriminator. I think the observation that the temperature control used in the Gumbel-softmax can reflect the trade-off between quality and diversity is interesting.

However, I feel the claim in the last sentence of the abstract and introduction is a little bit strong. Though this paper seems to be the first to really use Gumbel-softmax for text generation, similar techniques like using annealed softmax to approximate argmax has already been used in previous work (Zhang et al., 2017). Since this is similar to Gumbel-softmax, I think this may need additional one or two sentences to clarify this for more careful discussion.

Further, I would also recommend the authors discuss the following paper [a] to make this work more comprehensive as to the discussion of related work. [a] also uses annealed softmax approximation, and also divide the GAN approaches as RL-based and RL-free, similar in spirit as the discuss in this paper.

[a] Adversarial Text Generation via Feature-Mover's Distance, NIPS 2018.

(2) Presentation: This paper is carefully written and easy to follow. I enjoyed reading the paper.

(3) Evaluation: Experiments are generally well-executed, with ablation study also provided. However, human evaluation is lacked, which I think is essential for this line of work. I have a few questions listed below.

Questions:

(1) In section 2.4, it mentions that the generator needs pre-training. So, my question is: does the discriminator also need pre-training? If so, how the discriminator is pre-trained?

(2) In Table 1 & 2 & 3, how does your model compare with MaskGAN? If this can be provided, it would be better.

(3) Instead of using NLL_{gen}, a natural question is: what are the self-BLEU score results since it was used in previous work?

(4) The \beta_max value used in the synthetic and real datasets is quite different. For example, \beta_max = 1 or 2 in synthetic data, while \beta_max = 100 or 1000 is used in real data. What is the observation here? Can the authors provide some insights into this?

(5) I feel Figure 3 is interesting. As the authors noted, NLL_gen measures diversity, NLL_oracle measures quality. Looking at Figure 3, does this mean GAN model produces higher quality samples than MLE pretrained models, while GAN models also produces less diverse samples than MLE models? This is due to NLL_gen increases after pretraining, while NLL_oracle further decreases after pretraining. However, this conclusion also seems strange. Can the authors provide some discussion on this?

(6) Can human evaluation be performed since automatic metrics are not reliable enough?

---

> ### Author Response · Authors · 2018-11-14
> **Reply to Reviewer3 (Part 1)**
>
> Thank you very much for your review and helpful comments. We address your specific questions and comments below:
>
> *Related Work*
> 1. TextGAN: Thanks for the reviewer’s suggestion! We have provided a more detailed discussion in the “Related Work” Section to clarify the difference between RelGAN and TextGAN (Zhang et al., 2017) in dealing with the non-differentiability issue.
>
> 2. FM-GAN: Thanks for pointing out this recent paper. We have also added a discussion of this paper in the “Related Work” Section.
>
> *Questions*
> 1. “Does discriminator need pre-training?”
>
>  No, the discriminator in RelGAN does not need pre-training.
>
> 2. “How does RelGAN compared to MaskGAN?”
>
> [1] has recently showed that MaskGAN has much lower BLEU scores compared to our baseline models (MLE, SeqGAN, RankGAN, LeakGAN), where the evaluation settings in [1] are the same with our work. Thus, MaskGAN also performs worse than RelGAN in terms of BLEU scores.
>
> 3. “What are the self-BLEU score results since it was used in previous work?”
>
> A short answer: We did evaluate all methods using our implemented self-BLEU scores. However, we found that they may not be suitable for evaluating sample diversity. Also, we found an issue in the implementation of self-BLEU on the open-source Texygen platform that was used in prior work. We are currently in contact with authors of Texygen regarding the issue. Before we reach an agreement, we think it would be better not to use self-BLEU scores.
>
> More details:
> As far as we know, self-BLEU scores are first proposed by the authors of the Texygen benchmarking platform to evaluate diversity, where the basic idea is to calculate the BLEU scores by choosing each sentence in the set of generated sentences as hypothesis and the others as reference, and then take an average of BLEU scores over all the generated sentences. However, when looking into the implementation of self-BLEU scores: https://github.com/geek-ai/Texygen/blob/master/utils/metrics/SelfBleu.py, we found a severe issue inside for evaluating self-BLEU over training: Only in the first time of evaluation that the reference and hypothesis come from the same “test data” (i.e. the whole set of generated sentences). After that, the hypothesis keeps updated but the reference remains unchanged (due to “self.is_first=False”), which means hypothesis and reference are not from the same “test data” any more, and thus the scores obtained under this implementation is not self-BLEU scores!
>
> To this end, we modified their implementation to make sure that the hypothesis and reference are always from the same “test data” and found that the self-BLEU (2-5) scores are always 1 when evaluating all the models (MLE, SeqGAN, RankGAN, LeakGAN and RelGAN). Also as inspired by the Reviewer2’s comments, we tried to reduce the number of the “test data” by applying a small portion of the whole generated data as reference and still get 1 for the self-BLEU scores (even for the portion 5%). Therefore, if we understand correctly, self-BLEU scores may not be suitable for evaluating sample diversity.
>
> 4.  “Why is the \beta_max value used in the synthetic and real datasets quite different?”
>
> As there is a trade-off between sample diversity and quality in RelGAN, controlled by the tunable parameter \beta_max in both synthetic and real data, we can adjust \beta_max depending on different evaluation goals. In the synthetic data experiments, the goal is to show that RelGAN could outperform other models in terms of NLL_oracle (i.e. sample quality) regardless of the sample diversity. Thus, we chose very small \beta_max (1 or 2). In the real data experiments, however, the goal was to show that RelGAN could generate real text with both high quality and better diversity. Thus, we chose some intermediate values of \beta-max (100 or 1000) to find a good trade-off where both BLEU and NLL_gen scores can outperform other models. If instead we had chosen \beta_max=1, we would get significantly higher BLEU scores (as shown in Table 7 of Appendix H in the revised version of the paper), but at the cost of worse NLL_gen scores than other models (which would be in conflict with the main goal of the real data experiments).

---

> > ### Author Response · Authors · 2018-11-14
> > **Reply to Reviewer3 (Part 2)**
> >
> > 5. “Discuss the phenomenon -- NLL_gen increases after pre-training, while NLL_oracle decreases after pretraining in Figure 3?”
> >
> > This phenomenon is not unique in RelGAN (with a small \beta_max in particular). In general, GANs can generate text with better quality but are also more likely to suffer from mode collapse than the MLE trained counterparts on text generation, as has been shown in many previous works, such as [2,3,4]. This is why we have always seen a decrease in the NLL_oracle score (better sample quality) together with an increase of the NLL_gen score (worse sample diversity) while training current GANs (not only RelGAN with a small \beta_max) on the synthetic data. What makes RelGAN different from other GANs is that we can control the trade-off between sample quality and diversity with a single tunable hyperparameter. Tables 2 & 3 showed that if \beta_max is tuned properly, we can make the sample diversity of RelGAN to be very close to (or even better than) that of the MLE pre-trained LSTMs, while at the same time achieving much better sample quality.
> >
> > 6. “Can human evaluation be performed since automatic metrics are not reliable enough?”
> >
> > Thanks for the suggestion! We have done human evaluation for the EMNLP2017 WMT News dataset on Amazon Mechanical Turk, where the evaluation criteria details are provided in Table 5 (Appendix B.2 in the revised version of the paper) and the results are shown in Table 4 (Section 3.3 in the revised version of the paper). We can see the human scores in Table 4 also prefer RelGAN to other GANs and the MLE baseline models.
> >
> > Please let us know if we have addressed your concerns and if you have further comments.
> >
> >
> > [1] Lu et al., “Neural Text Generation: Past, Present and Beyond.” arXiv preprint arXiv:1803.07133.
> > [2] Fedus et al., “MaskGAN: Better Text Generation via Filling in the _.”  in ICLR 2018.
> > [3] Zhu et al., “Texygen: A Benchmarking Platform for Text Generation Models.” in SIGIR 2018.
> > [4] Semeniuta et al., “On accurate evaluation of gans for language generation.” arXiv preprint arXiv:1806.04936, 2018.

---

### Official Review · AnonReviewer2 · 2018-11-02
**Good Paper**

**Rating:** 8
**Confidence:** 4

**Review:**

Update: the authors' response and changes to the paper properly addressed the concerns below. Therefore the score was improved from 6 to 8.

----


The paper makes several contributions: 1. it extends GAN to text via Gumbel-softmax relaxation, which seems more effective than the other approaches using REINFORCE or maximum likelihood principle. 2. It shows that using relational memory for LSTM gives better results. 3. Ablation study on the necessity of the relational memory, the relaxation parameter and multi-embedding in the discriminator is performed.

The paper's ideas are novel and good in general, and would make a good contribution to ICLR 2019. However, there are a few things in need of improvement before it is suite for publication. I am willing to improve the scores if the following comments are properly addressed.

First of all, the paper does not compare with recurrent networks trained using only the "teacher-forcing" algorithm without using GAN. This means that at a high level, the paper is insufficient to show that GAN is necessary for text generation at all. That said, since almost every other text GAN paper also failed to do this, and the paper's contribution on using Gumbel-softmax relaxation and the relational memory is novel, I did not get too harsh on the scoring because of this.

Secondly, whether using BLEU on the entire testing dataset is a good idea for benchmarking is controversial. If the testing data is too large, it could be easily saturated. On the other hand, if the testing data is small, it may not be sufficient to capture the quality well. I did not hold the authors responsible on this either, because it was used in previously published results. However, the paper did propose to use an oracle, and it might be a good idea to use a "teacher-forcing" trained RNN anyways since it is necessary to show whether GAN is a good idea for text generation to begin with (see the previous comment).

A third comment is that I had wished the paper did more exploration on the relaxation parameter \beta. Ideally, if \beta is too large, the output would be too skewed towards a one-hot vector such that instability in the gradients occurs. On the other hand, if \beta is too small, the output might not be close enough to one-hot vectors to make the discriminator focus on textual differences rather than numerical differences (i.e., between a continuous and a one-hot vector). It would make sense for the paper to show both ends of these failing cases, which is not apparent with only 2 hyper-parameter choices.

Finally, the first paragraph in section 2.2.2 suggests that the gap between discrete and continuous outputs is the reason for mode collapsing. This is false. For image generation, when all the outputs are continuous, there is still mode collapsing happening with GANs. The authors could say that the discrete-continuous gap contributes to mode-collapsing, but this is not too good either because it will require the paper to conduct experiments beyond text generation to show this. Authors should make changes here.

---

> ### Author Response · Authors · 2018-11-14
> **Reply to Reviewer2 (Part 1)**
>
> Thank you very much for your review and helpful comments. We address your specific questions and comments below:
>
> 1. “The paper does not compare with RNNs trained using only the "teacher-forcing" algorithm without using GAN.”
>
> We think there are some misunderstandings here that we would like to clarify. One of our baseline algorithms is  “MLE”,  where recurrent networks are trained by using the teacher-forcing algorithm (see Tables 1, 2 & 3 for comparison with RelGAN). RelGAN consistently outperforms the baseline model “MLE” in terms of both BLEU scores (sample quality) and NLL_gen (sample diversity). For clarity, we have added a sentence in Section 3.1 to explicitly explain “MLE” in the revised version of the paper.
>
> 2. “Whether using BLEU on the entire testing dataset is a good idea for benchmarking is controversial.”
>
> First, we agree that absolute BLEU scores depend on the size of test dataset. As such, we have used different subsets (25%, 50%, 75%, 100%) of the original test data for COCO Image Captions to evaluate the generated text from both RelGAN and MLE. We evaluate each subset of the test data 6 times and record the average BLEU scores. We find that for both RelGAN and MLE, the (average) BLEU scores consistently increase with the fraction of test data used. Therefore, whenever we show BLEU scores, the size of the test data MUST be provided as well for fair comparison. This is similar to the Frechet inception distance (FID) score used for evaluating generated images, where the number of generated and real samples used to calculate the FID score also influences the value of the score.
>
> Second, the following empirical observation shows that the reviewer’s concern on the “controversiality of BLEU scores” may be not that severe in this paper: The relative BLEU score differences between RelGAN and MLE remain approximately invariant for different portions of the test data. For instance, BLEU-2 with portion 25% is 0.750 for RelGAN vs. 0.649 for MLE (with the difference 0.101) and BLEU-2 with portion 75% is 0.811 for RelGAN vs. 0.718 for MLE (with the difference 0.093). Thus, if we focus on the relative comparison between different models, the size of the test dataset may not matter much, as long as it is not too small or too large.
>
> Finally, if we correctly understand the reviewer's suggestion of using a “teacher-forcing” trained RNN for evaluation, the reviewer refers to the “validation perplexity” metric. [1] has previously used validation perplexity to evaluate the sample quality of MaskGAN, where they showed that in some cases, the perplexity increases steadily while sample quality still remained relatively consistent. More broadly, [2] has also pointed out that validation perplexity does not necessarily correlate with sample quality in the evaluation of generative models, and so validation perplexity may not be a good replacement for BLEU scores in terms of evaluating sample quality. Instead, as suggested by Reviewer3, we have added the human Turing test on Amazon Mechanical Turk (where results are shown in Table 4 in the revised version of the paper), which we think could be a good complementary to BLEU scores.
>
>
> [1] Fedus et al., “MaskGAN: Better Text Generation via Filling in the _.”  in ICLR 2018.
> [2] Theis et al., “A note on the evaluation of generative models.” in ICLR 2016.

---

> > ### Author Response · Authors · 2018-11-14
> > **Reply to Reviewer2 (Part 2)**
> >
> > 3. “It would make sense for the paper to show both ends of these failing cases with the exploration on more values of \beta_max.”
> >
> > Thank you for the suggestion. We have added Appendix H to explore the impact of different inverse temperature \beta_max in RelGAN, especially where the failing cases at the two extremes are discussed.
> >
> > First, our results confirm that, similar to the synthetic data experiments, there also exists a consistent trade-off between sample quality and diversity in the real data, controlled by the tunable hyperparameter \beta_max. It also reveals failing cases at the two extremes: On the one hand, if \beta_max is too small, RelGAN suffers from severe mode collapse (large NLL_gen scores) and training instability (high variance of scores). On the other hand, if \beta_max is too large, the sample quality improvement of RelGAN becomes marginal (low BLEU scores). Therefore, in the main text we have chosen the two intermediate values, \beta_max = 100 & 1000, to show the advantages of RelGAN over other models, while still demonstrating its ability to control the trade-off between sample quality and diversity.
> >
> > 4. “The first paragraph in section 2.2.2 in terms of describing mode collapse is misleading.”
> >
> > We agree that the last sentence of the first paragraph in Section 2.2.2 is a little bit misleading. We appreciate the reviewer’s suggestion and have changed it to “Intuitively, this might be one factor that contributes to mode collapse in RelGAN on text generation.”, which we believe avoids making an argument that applies to mode collapse in general GANs on image generation.
> >
> > Please let us know if we have addressed your concerns and if you have further comments.

---

> > ### Comment · AnonReviewer2 · 2018-11-14
> > **Good Improvement**
> >
> > The changes addressed my concerns and the score is now improved from 6 to 8.
> >
> > For "teacher-forcing": I misread MLE as the same as MaliGAN, given the name similarities ("maximum likelihood"). Thanks for the clarification.

---

### Official Review · AnonReviewer1 · 2018-11-05
**Interesting work which makes Gumbel-softmax relaxation work in GAN-based text generation using a relational memory**

**Rating:** 6
**Confidence:** 4

**Review:**

Overall:
This paper proposes RelGAN, a new GAN architecture for text generation, consisting of three main components: a relational memory based generator for the long-distance dependency modeling, the Gumbel-Softmax relaxation for training GANs on discrete data, and multiple embedded representations in the discriminator to provide a more informative signal
for the generator updates.

Quality and Clarity:
The paper is well-written and easy to read.

Originality :
Although each of the components (relational memory, Gumbel-softmax) was already proposed by previous works, it is interesting to combine these into a new GAN-based text generator.
However, the basic setup is not novel enough. The model still requires pre-training the generator using MLE. The major difference are the architectures (relational memory, multi-embedding discriminator) and training directly through Gumbel-softmax trick which has been investigated in (Kusner and Hernandez-Lobato, 2016).

Significance:
The experiments in both synthetic and real data are in detail, and the results are good and significant.

-------------------
Comments:
-- In (4), sampling is known as non-differentiable which means that we cannot get a valid definition of gradients. It is different to denote the gradient as 0.
-- Are the multiple representations in discriminator simply multiple “Embedding” matrices?
-- Curves using Gumbel-softmax trick + RM will eventually fall after around 1000 iterations in all the figures. Why this would happen?
-- Do you try training from scratch without pre-training? For instance, using WGAN as the discriminator


Related work:
-- Maybe also consider to the following paper which used Gumbel-softmax relaxation for improving the generation quality in neural machine translation related?
Gu, Jiatao, Daniel Jiwoong Im, and Victor OK Li. "Neural machine translation with gumbel-greedy decoding." arXiv preprint arXiv:1706.07518 (2017).

---

> ### Author Response · Authors · 2018-11-14
> **Reply to Reviewer1**
>
> Thank you very much for your review and helpful comments. We address your specific questions and comments below:
>
> 1. “Non-differentiability is different to denote the gradient as 0.”
>
> We agree that in general non-differentiability does not directly imply a vanishing gradient as in (4). To draw this conclusion, we first consider that the sampling operations in (3) are not differentiable, i.e., the output of the generator is discrete, taking values from a finite set. This implies a step function (with multiple steps in general) at the end of the generator, which is not differentiable only at a finite set of points (with measure zero). Since the derivative of a step function is 0 almost everywhere, the gradient of the generator’s output w.r.t. its parameters will also be zero almost everywhere. For clarity, we have added this reasoning to the revised version of the paper.
>
> 2. “Are the multiple representations in discriminator simply multiple “Embedding” matrices?”
>
> Yes, we apply multiple different embedding matrices, each of which linearly transforms one input sentence into a separate embedded representation. In our proposed discriminator framework, each embedded representation is independently passed through the later layers of the discriminator neural network (denoted as “CNN-based classifier” in the paper) and the loss function to obtain an individual score (e.g. “real” or “fake”). Finally, the ultimate score to be propagated back to the generator is the average of these individual scores. Our ablation study experiments showed the advantages of this simple improvement in the discriminator.
>
> 3. “Why curves in RelGAN eventually fall after around 1000 iterations?”
>
> Good point! We have added Appendix F in the revised version of the paper to discuss this phenomenon. As shown in Figure 10, there is a diversity-quality transition during training of RelGAN: Early on in training, it learns to aggressively improve sample quality while sacrificing diversity. Later on, it turns instead to maximizing sample diversity while gradually decreasing sample quality. Intuitively, we think it may be much easier for the generator to produce realistic samples -- regardless of their diversity -- in order to fool the discriminator in the early stages of training. As the discriminator becomes better at distinguishing samples with less diversity over iterations, the generator has to focus more on producing more diverse samples to fool the discriminator.
>
> 4. “Do you try training from scratch without pre-training?”
>
> Yes, we have tested the performance of RelGAN without pre-training by using different GAN losses (including WGAN-GP as the reviewer has mentioned). The results and analysis are provided in Appendix G. We find that without pre-training, there is still a significant improvement for RelGAN compared with the random generation, in particular for the standard GAN loss. In contrast, without pre-training, previous RL-based GANs for text generation, such as SeqGAN and RankGAN, always get stuck around their initialization points and are not able to improve performance at all. This demonstrates that RelGAN may be a more promising GAN architecture to explore in order to reduce dependence on pre-training for text generation. In future work, we plan to do an extensive hyperparameter search to further improve the performance of RelGAN without pre-training.
>
> Related Work: Thanks for pointing out this paper. We have added a discussion of this paper in the “Related Work” Section.
>
> Please let us know if we have addressed your concerns and if you have further comments.

---

### Meta-Review · Area_Chair1 · 2018-11-06
**Good paper; a little incremental**

**Confidence:** 3
**Recommendation:** Accept (Poster)

**Metareview:**


pros:
- well-written and clear
- good evaluation with convincing ablations
- moderately novel

cons:
- Reviewers 1 and 3 feel the paper is somewhat incremental over previous work, combining previously proposed ideas.

(Reviewer 2 originally had concerns about the testing methodology but feels that the paper has improved in revision)
(Reviewer 3 suggests an additional comparison to related work which was addressed in revision)

I appreciate the authors' revisions and engagement during the discussion period.  Overall the paper is good and I'm recommending acceptance.